# Non-Surgical Management of Upper Cervical Instability via Improved Cervical Lordosis: A Case Series of Adult Patients

**DOI:** 10.3390/jcm12051797

**Published:** 2023-02-23

**Authors:** Evan A. Katz, Seana B. Katz, Michael D. Freeman

**Affiliations:** 1Independent Researcher, Boulder, CO 80302, USA; 2Faculty of Health Medicine and Life Sciences, Maastricht University, 6229 ER Maastricht, The Netherlands

**Keywords:** cervical lordosis, motor vehicle crash, digital motion X-ray, upper cervical instability

## Abstract

Injury to the head and neck resulting from whiplash trauma can result in upper cervical instability (UCIS), in which excessive movement at C1 on C2 is observed radiologically. In some cases of UCIS there is also a loss of normal cervical lordosis. We postulate that improvement or restoration of the normal mid to lower cervical lordosis in patients with UCIS can improve the biomechanical function of the upper cervical spine, and thus potentially improve symptoms and radiographic findings associated with UCIS. Nine patients with both radiographically confirmed UCIS and loss of cervical lordosis underwent a chiropractic treatment regimen directed primarily at the restoration of the normal cervical lordotic curve. In all nine cases, significant improvements in radiographic indicators of both cervical lordosis and UCIS were observed, along with symptomatic and functional improvement. Statistical analysis of the radiographic data revealed a significant correlation (R^2^ = 0.46, *p* = 0.04) between improved cervical lordosis and reduction in measurable instability, determined by C1 lateral mass overhang on C2 with lateral flexion. These observations suggest that enhancing cervical lordosis can contribute to improvement in signs and symptoms of upper cervical instability secondary to traumatic injury.

## 1. Introduction

Whiplash is an injury mechanism most typically associated with the rapid flexion/extension, compression and rotation of the cervical spine that can occur in a motor-vehicle crash [1]. Injuries resulting from whiplash trauma are common; it is estimated that there are approximately 2.9 million cases of whiplash trauma-associated injury that occur annually in the US [2]. The constellation of chronic symptoms, largely affecting the head and neck, that can result from acute injury after whiplash trauma can present a complex problem for both patients and clinicians [3]. Chronic symptoms associated with “late whiplash” can include headaches, dizziness, neck and upper back pain, as well as widespread pain [4]. Cervical spine pathology associated with whiplash trauma includes facet derangement, disk injury, and spinal ligament strain and rupture, often in the upper cervical spine [5].

Whiplash trauma can also result in upper cervical instability (UCIS), a condition in which excessive movement is observed at the C1–2 levels in combination with a wide constellation of head and neck somatic signs and symptoms [6,7].

UCIS is typically identified and diagnosed by comparing radiographic findings in patients with clinical complaints to accepted normal radiographic values (Figure 1). Radiographic evidence of UCIS includes anterior translation of C1 on C2 such that the atlanto-dental interspace exceeds 3.5 mm (as observed on lateral flexion radiographs), [8] and lateral translation of C1 on C2 such that there is more than 2.0 mm of lateral overhang of the lateral mass of C1 on the superior articulating facet of C2 (as observed in anterior to posterior (AP) open-mouth radiographs with lateral bending movements), in combination with asymmetry of the peri-odontoid space. While the 2.0 mm overhang threshold has only moderate sensitivity and positive predictive value (PPV) for upper cervical injury in the whiplash-injured population with chronic symptoms (64% and 75%, respectively), and the more subjective asymmetry assessment has low sensitivity (29%) but high PPV (95%), in combination the two findings have a PPV of 100% [9]. 

The presence of instability is generally assumed to result from ligamentous and facet capsule damage resulting from the incipient trauma [10]. The loss of ligamentous integrity in the upper cervical spine in turn raises the concern of increased risk of future injury in the unfortunate event of a subsequent trauma [11]. Patients with symptomatic UCIS may complain of symptoms with varying degrees of specificity to the pathology—ranging from seemingly high specificity (i.e., difficulty holding the head up without support, intolerance to prolonged static postures, persisting sensation of suboccipital clicking) to nonspecific (i.e., head, neck, and shoulder pain) [12,13].

For UCIS patients with refractory symptoms, surgical fusion of the C1–C2 vertebrae is a viable, albeit under-investigated therapeutic option, as success and complication rates for the relatively rare procedure are not reliably established in the literature. Like all spinal fusion surgeries, fusion for C1–2 instability is expensive, invasive, and carries some degree of risk [14]. In addition to the immediate risks associated with surgery such as blood loss and neurological injury, intermediate (i.e., infection, graft subsidence) and long-term risks (adjacent segment pathology) are also potential complications of spinal fusion surgery [15]. In spite of the risks, upper cervical fusion is often the only option presented to the patient with refractory symptoms and demonstrable UCIS. 

Loss of the normal cervical lordotic curve is a common radiographic finding in patients with chronic pain after whiplash [16], although there is no general consensus in the literature as to whether the finding indicates true pathology or a normal variant [17,18,19,20]. It is well established, however, that the normal cervical lordosis is the biomechanically ideal posture of the cervical spine, as mechanical stresses in the spine are most evenly balanced between the intervertebral disk and zygapophyseal joints when the “C”-shaped curve of the neck is maintained [21]. The clinical benefits of a lordotic cervical curve have been demonstrated in multiple studies. As an example, in a study of 300 neck pain patients under the age of 40, Gao and colleagues found an increased degree of disk herniation in the patients with straight and kyphotic cervical spines, in comparison with the lordotic necks [22]. They also reported an improvement in disk height and a decrease in disk herniation severity and associated spinal cord compression in the patients who had an improvement in lordosis. A recent systematic review of controlled clinical trials of lordosis restoration therapy for neck pain patients demonstrated that when treatment included extension traction directed at improvement of the lordotic curve, symptomatic improvements were maintained for more than 1 year after cessation of therapy [23]. In comparison, control treatment groups without extension traction were more likely to relapse after cessation of therapy. 

A therapeutic model directed at methods of restoring the normal cervical curve is called Chiropractic Biophysics^TM^ (CBP). CBP relies on a combination of common chiropractic modalities (e.g., manipulation), Mirror Image^®^ exercises, and spinal extension traction (Figure 2) [24]. There is evidence that suggests that CBP therapy is effective for restoring cervical lordosis [25].

While it is the mid to lower cervical spine that benefits most from the restoration of normal lordosis, there is evidence that the upper cervical spine can also benefit from a normal cervical curve, as a straight or kyphotic cervical spine is compensated at the C0–1–2 level by excessive craniocervical extension in an effort to keep the eyes level with the horizon [26,27]. 

In the present investigation, we describe nine cases of radiographically confirmed and symptomatically congruent UCIS in patients with chronic symptoms following whiplash trauma. In all nine cases, the patients were also found to have a reduction in normal cervical lordosis, and thus treatment was directed at restoring the lordotic curve via the CBP^®^ approach. Baseline and post-treatment radiographic parameters of both UCIS and cervical lordosis are described, as well as subjective response to treatment. 

## 2. Materials and Methods

This case series includes nine patients (2 male, 7 female), ranging in age from 28 to 52 years with an average age of 39 years (Table 1). Each patient presented to the same chiropractic practice (authors EK and SK) for evaluation of acute or chronic symptoms consistent with upper cervical instability. The majority of the patients had undergone evaluation with other clinicians, including neurosurgeons or orthopedic spine surgeons, or had been previously treated with physical therapy or chiropractic manipulation. The inclusion criterion for the cases was all consecutive patients presenting with radiographic evidence of both UCIS and loss of cervical lordosis, following a history of traumatic injury of the neck (primarily whiplash trauma). A finding of fracture, dislocation, or myelopathy or other concerning neurological manifestation of the instability was an exclusion criterion, as such patients would be uniformly referred for urgent neurosurgical or orthopedic evaluation as part of the clinic protocol. The patient histories and treatment course varied widely, and the median time between baseline and follow-up radiographic examination was 16 weeks (with an interquartile range of 32 weeks). 

### 2.1. Radiographic Analysis 

Video-Fluoroscopic (VF) examination of the cervical spine was performed using digital motion X-ray (DMX). This imaging protocol allows for continuous examination of movement within the cervical spine. DMX records 30 images per second of continuous X-ray and captures an active range of motion allowing dynamic four-dimensional visualization of the integrity of the ligaments of the upper cervical spine. DMX imaging, therefore, provides the opportunity to assess both static and dynamic parameters of vertebral alignment [28,29]. 

Two DMX views were used to assess the degree of lordotic curvature and to identify and quantify findings consistent with UCIS; a neutral lateral cervical (NLC) and anterior to posterior open-mouth lateral cervical bending (APOM-LCB). Both examinations were performed at the baseline and prior to initiation of therapy, and then repeated no less than 72 h after therapy was concluded, as the goal was to avoid imaging of any temporary cervical curve improvement directly following extension traction. In order to produce images that were consistent with each other, the patient was positioned in the same fashion in both studies, each conducted by the same author, (either EAK or SBK), with the central ray at C5, back or shoulder touching the image intensifier (depending on view), and with a 20 mm marker on the patient’s skin for calibration of the PostureRay^®^ measuring software.

Actual Rotational Angles (ARA) were calculated from sagittal NLC images using PostureRay^®^ software (PostureCo, Inc., Trinity, FL, USA) for Computerized Radiographic Mensuration Analysis (see Figure 1a). The cervical ARA is the angular measurement between the posterior vertebral body margins of C2 and C7, and the average ARA for a maintained cervical lordosis is −34° [30]. All images include a standard X-ray marker for calibration prior to each measurement in order to avoid magnification error.

The ARA was used to quantify the deviation of the segmental rotational angles from C2–C7 from normal cervical lordosis values. Static images of right and left APOM-LCB were taken as frames from DMX videos at the extremes of comfortable lateral flexion. The images were analyzed using the PostureRay^®^ software to quantify the amount of C1–C2 lateral mass overhang margin at maximum right and left lateral cervical bending (Figure 1b). An overhang margin of >3 mm was used as the threshold for the study inclusion criterion of potential C1 on C2 instability, in combination with asymmetry of the peri-odontoid space. As noted above, at this threshold of combined findings, the sensitivity (i.e., true positive rate) and positive predictive value (i.e., true positive rate/all positives) for traumatic injury is 100% [9]. 

Along with findings consistent with UCIS, included patients also demonstrated a loss of lordosis, defined as an increase from the average normal lordotic ARA of −34° (see Table 2), resulting in an appearance of straightening or reversal (i.e., kyphosis) of the normal lordotic curve. Combined with an initial evaluation indicating symptomatic instability, the radiological examination confirmed a diagnosis of both loss of normal cervical lordosis and upper cervical instability for each of the nine patients included for study, as well as some degree of presumed injury to the upper cervical ligaments, including the alar, transverse, and other stabilizing ligaments [31].

### 2.2. Intervention 

Patients were treated twice per week on average for the indicated durations of treatment between radiographic evaluations (Table 1). Treatments incorporated full spine chiropractic adjustments, as well as Mirror Image^®^ adjustments using a drop-piece table. Mirror Image^®^ adjustments involve placing the cervical spine into an extended, overcorrected position during the chiropractic adjustment in order to achieve optimal progression toward proper spinal alignment [32]. The manipulations were solely directed at hypomobile spinal segments in the mid and lower cervical spine, as manipulation at the unstable upper cervical spine would be contraindicated. Several forms of cervical extension traction to restore or improve the cervical lordosis were also administered. These consisted of the following:(1)Use of a Cervical posture pump^®^ (Posture Pro, Inc., Huntington Beach, CA, USA), a self-controlled device with an inflatable airbladder that is applied to the supine mid-cervical spine. See Figure 2a.(2)Home use of a cervical Denneroll™ (Denneroll Industries International Pty Ltd., Sydney, Australia), used like a pillow while the patient is supine, and positioned at the mid to lower cervical spine. See Figure 2b.(3)Once tolerance to the previous two devices was established, the patient was progressed into a form of 2-way extension traction performed in office [33]. This therapy is applied while the patient lies supine on a specially designed chair, that employs a forehead harness to fix the head in a slightly extended position. A second strap is used to apply anterior tension to the mid to lower cervical spine, along the plane of the mid cervical spinal disks. See Figure 2c.

### 2.3. Statistical Analysis

Average treatment effect was assessed via the difference between the pre- and post-treatment radiographic measurements of ARA and lateral mass overhang using the Student’s t-test for normally distributed differences and the signed rank test for non-normally distributed differences. Linear regression was used to assess the correlation between the percent change in ARA and the average of the left and right percent changes in C1–C2 overhang measurement (percent change = [post-measurement − pre-measurement]/pre-measurement; average overhang percent change = [left percent change + right percent change]/2). Normality of each difference and the average overhang percent change was assessed using the Shapiro-Wilk test. *p*-values < 0.05 were considered significant. All analyses were performed using SAS Software, version 9.4 (SAS Institute, Inc., Cary, NC, USA).

## 3. Results

Following the intervention period, clinical evaluations and radiographic analyses of each patient were repeated. After the intervention, each patient described marked improvements in overall pain scores, cervical range of motion, and quality of life. Patients who reported symptoms most closely associated with UCIS, including dizziness and blurred vision (patients one, two, five, and six), reported cessation that they are no longer experiencing those symptoms as of the end of this study. Additionally, those patients who had been managing their pain with prescription pain medications were no longer doing so. Furthermore, each patient described in this report has been able to resume activities which had been precluded by their neck pain and symptoms relating to instability. Patient three was able to resume participation in martial arts, and patients four and five also reported improved function. Patients seven, eight and nine reported a decrease or cessation of chronic and frequent headaches. 

Radiographic re-evaluations, performed at least 72 h after the most recent therapy, revealed substantially improved cervical lordosis (i.e., progress toward the ideal ARA of −34°) in all of the patients. Mean ARA value at baseline was −10°, compared to −21° after the intervention (*p* = 0.002, see Table 3). Three of the patients (one, six, and nine) had ARA values at or approaching −30° (see Table 2). There was an average reduction in C1–C2 lateral mass overhang from 6.0 mm to 3.0 mm on the left (*p* = <0.001), and from 3.1 mm to 1.6 mm on the right (*p* = 0.004) (see Table 3). The average percent change in C1–2 overhang was normally distributed (*p* = 0.91). The percent change in ARA and average percent change in C1–2 overhang were moderately correlated (see Figure 3, R^2^ = 0.46, *p* = 0.04).

## 4. Discussion

There are several plausible explanations for the observed association between symptomatic improvement, improved cervical lordosis, and decreased C1–2 instability in the described cases. One explanation is that the symptoms resolved spontaneously, and that the improvements were unrelated to the treatment or radiographic changes. While plausible, this explanation defies logic and convention. The patients had been symptomatic for months to years and had all tried other treatments without success prior to initiating the cervical lordosis correction therapy. The positive changes observed in the imaging are thus much more likely to be explained by the therapy, rather than the natural course of the condition, which had reached a static level in all of the patients.

The remaining explanations are that the therapy directed at improving cervical lordosis improved the lordosis, the symptoms, and the C1–2 instability, or that the symptoms and instability improved for some reason unrelated to the alteration of the lordosis. We favor the former explanation. The upper cervical instability (and associated symptoms indicative of UCIS) is the result of upper cervical ligament injury and associated laxity. Loss of normal cervical lordosis produces a relatively flexed posture of the upper cervical spine, requiring extension accommodation at the head to keep the neutral gaze level with the horizon [34,35]. It makes sense that a persistently abnormal posture of the upper cervical spine would likely put a higher degree of strain on the upper cervical ligaments during normal activity, relative to having the head in a neutral position relative to C1–2, as occurs with normal extension at the craniocervical junction. We hypothesize that improvement of the cervical lordosis results in improved biomechanics of the upper cervical spine, and that this in turn allows for improvement of the integrity of the ligaments responsible for craniocervical stability. This hypothesis is an extension of the findings of prior authors, who have described a correlation between increased angle of the upper cervical (C0–2) spine and increased risk of cervical kyphosis [27]. Ours is the first study to demonstrate a relationship between loss of normal cervical curve and symptomatic instability, however.

Because the design of the present study was conceived of only after the association between cervical curve improvement and decreased upper cervical instability was noted, the evidence for symptomatic improvement was derived from narrative histories, rather than consistently used metrics. Future investigation would thus benefit from an a priori design with standardized objective measurements of the non-radiographic changes described in this study (e.g., Neck Disability Index, etc.), as well as the inclusion of a comparison group of patients who did not improve radiographically in either cervical curve or upper cervical instability. Moreover, the ability to generalize from this small sample of highly selected patients is limited, and thus another goal for future investigation is to increase the number of study subjects.

## Figures and Tables

**Figure 1 jcm-12-01797-f001:**
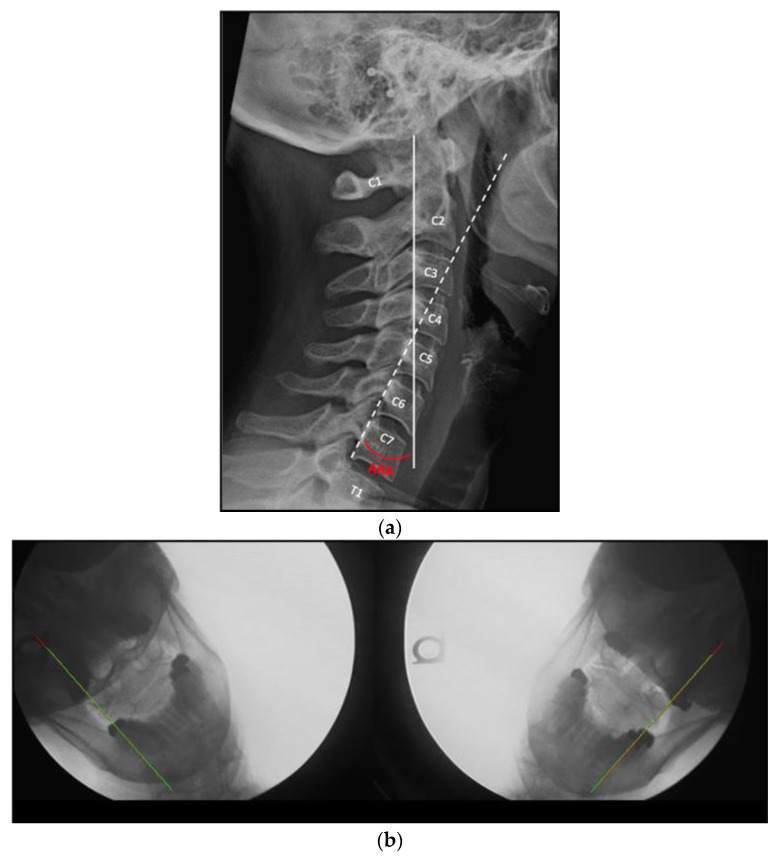
(**a**) Actual rotational angles (ARA) measurement in a patient with a normal cervical lordosis. The ARA (in red) indicates the angle between the posterior body margins of C2 (solid white line) and C7 (dashed white line). (**b**) AP open mouth lateral bending still shots from the DMX study (left and right lateral flexion on the left and right, respectively), with the green line indicating the lateral mass margin of C1, and the red line indicating the lateral body margin of C2. The lines overlap, indicating no overhang of C1 on C2.

**Figure 2 jcm-12-01797-f002:**
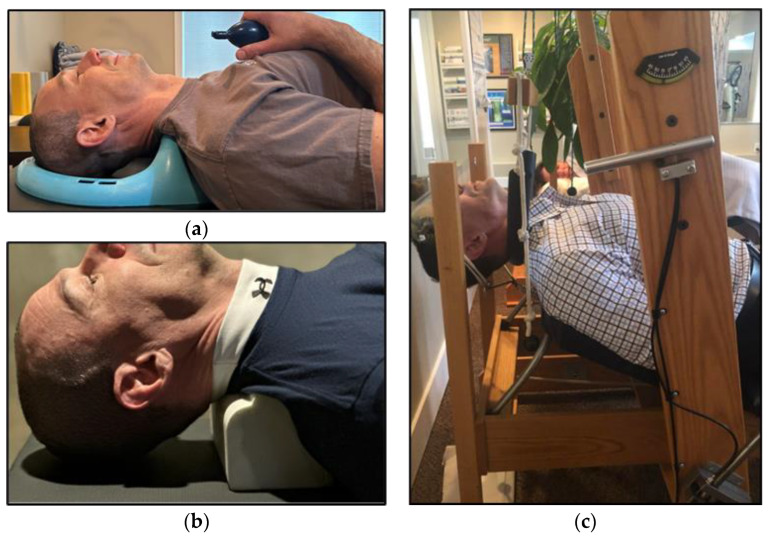
(**a**) Cervical posture pump^®^ demonstration. (**b**) Denneroll™ demonstration. (**c**) 2-way extension traction demonstration. Note: the model in all figures is author EAK.

**Figure 3 jcm-12-01797-f003:**
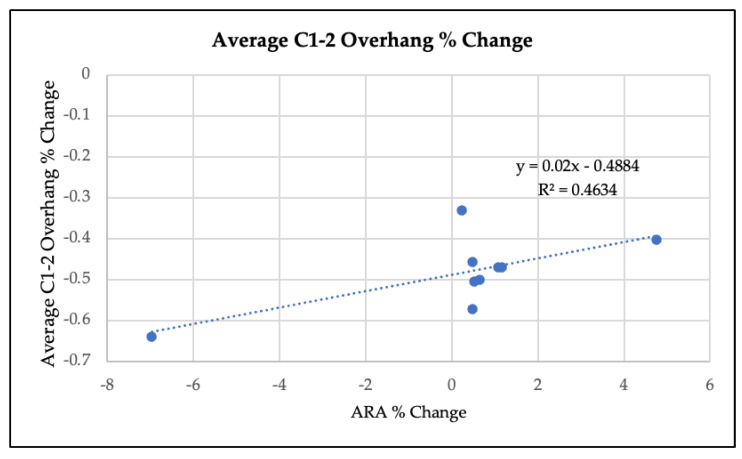
Linear regression analysis of average % change in C1-2 overhang versus the % change in ARA.

**Table 1 jcm-12-01797-t001:** Patient characteristics.

Patient Number	Gender (M/F)	Age	Duration ^a^ (Weeks)	Symptoms ^b^
1	F	46	7	Neck pain and weakness, dizziness.
2	F	36	10	Neck pain, headaches.
3	M	37	12	Pain at the base of the skull, clicking sensation dizziness.
4	F	52	16	Head and neck pain, blurred vision.
5	F	34	16	Head and neck pain, blurred vision, arm tingling, clicking sensation, sleep disruption.
6	M	28	19	Head and neck pain, occipital spasms, blurred vision, dizziness.
7	F	35	44	Head and neck pain, arm tingling.
8	F	53	52	Head and neck pain, head pressure, pain behind left eye, sleep disruption
9	F	30	68	Debilitating Headaches, neck pain.

^a^ Duration indicates the period of time between baseline radiographic measurement and follow-up evaluation. ^b^ Symptoms listed are only those consistent with UCIS. Patients may have had other less UCIS-specific symptoms.

**Table 2 jcm-12-01797-t002:** Radiographic measurements, pre and post intervention.

	Time of X-ray, Relative to Intervention ^a^	ARA ^b^ C2–C7	C1–2 Lateral Overhang Margin ^c^	
			Left	Right
Patient 1	Baseline	−14.1°	8.8 mm	6.1 mm
	Post intervention	−30.4°	6.3 mm	2.1 mm
Patient 2	Baseline	−4.1°	7.5 mm	1.8 mm
	Post intervention	−23.6°	2.3 mm	1.6 mm
Patient 3	Baseline	−2.8°	5.2 mm	2.4 mm
	Post Intervention	−4.6°	1.3 mm	1.8 mm
Patient 4	Baseline	3.0°	3.6 mm	1.2 mm
	Post intervention	−17.9°	1.1 mm	0.5 mm
Patient 5	Baseline	−8.8°	8.8 mm	6.1 mm
	Post intervention	−18.2°	6.3 mm	2.1 mm
Patient 6	Baseline	−19.9°	3.2 mm	3.0 mm
	Post intervention	−29.6°	2.2 mm	0.5 mm
Patient 7	Baseline	−11.2°	7.0 mm	1.9 mm
	Post intervention	−17.0°	2.5 mm	1.2 mm
Patient 8	Baseline	−12.0°	5.5 mm	3.5 mm
	Post intervention	−14.8°	2.8 mm	2.9 mm
Patient 9	Baseline	−19.7°	4.6 mm	2.0 mm
	Post intervention	−29.0°	2.0 mm	1.3 mm

^a^ During the period of time between the first and follow-up X-rays (the ‘treatment duration’), patients underwent treatment according to the intervention protocol. ^b^ ARA: Absolute Rotational Angle. Normal value is −34.0° or less. ^c^ Normal value 2 mm or less.

**Table 3 jcm-12-01797-t003:** Differences between post- and pre-treatment measurements (negative values denote improvement).

Patient	ARA	Left Overhang	Right Overhang
1	−16.3	−2.5	−4.0
2	−19.5	−5.2	−0.2
3	−1.8	−3.9	−0.6
4	−20.9	−2.5	−0.7
5	−9.4	−2.5	−4.0
6	−9.7	−1.0	−2.5
7	−5.8	−4.5	−0.7
8	−2.8	−2.7	−0.6
9	−9.3	−2.6	−0.7
Shapiro-Wilk test of normality *p*-value ^a^	0.4	0.29	0.004
Mean difference (standard deviation)	−10.6 (6.9)	−3.0 (1.27)	−1.6 (1.5)
Mean difference 95% CI	[−15.9, −5.3]	[−4.0, −2.1]	[−2.7, −0.4]
Test statistic ^b^	−4.6	−7.2	−22.5
Degrees of freedom (df)	8	8	NA
*p*-value	0.0018	<0.0001	0.0039

^a^ For sample sizes < 2000, Shapiro-Wilk is the appropriate test of normality. ^b^ Paired *t*-test for ARA and left overhang; signed rank test for right overhang.

## Data Availability

All data were provided in the manuscript.

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
