# Peer review of "Non-Surgical Management of Upper Cervical Instability via Improved Cervical Lordosis: A Case Series of Adult Patients"

_jcm, 2023, doi:10.3390/jcm12051797_

Round 1
Reviewer 1 Report
Thank you for the opportunity to review this manuscript. .
My edits and suggestions are provided below:
Comment 1: Overall Manuscript
Please consider reporting this case series in accordance with the CARE guidelines:
https://www.equator-network.org/reporting-guidelines/care/
This will require a change in the way that several sections are written and reported.
Comment 2: Title
Please indicate in the title of the manuscript that this is a case series involving adult patients.
Comment 3: Introduction
Please provide the reader with prevalence or incidence figures for both whiplash injury and upper cervical instability. This will allow the reader to gauge the magnitude of the issue. It might also be helpful to highlight the epidemiology regarding how many cases of radiographically appreciable instability are symptomatic. This will also comment on the significance of this work.
Comment 4:
Introduction, Page 2, Line 62. The importance of a normal cervical lordosis needs to be given more support in this section. Or at least, highlight that there are questions that we do not know the answers to at this point in time.
Is there evidence to suggest that it is clinically worthwhile restoring a theoretically ideal posture? Are the benefits worth the cost/effort? Asymptomatic patients often display a loss of the normal cervical lordosis, so is having a normal cervical lordosis essential from a clinical perspective?
Comment 5:
Introduction, Page 2, Line 72. CPB should read CBP.
Comment 6:
Materials and Methods
Page 2, Line 92. Please report median and interquartile range instead of mean the for the treatment times as these data are right (positively) skewed.
Comment 7:
Page 3 Line 98. 'Typo: less' instead of 'les'
Comment 8:
Have these patient's been referred for an orthopaedic assessment given that they have displayed radiographic and symptomatic instability in the upper cervical spine?
Comment 9:
Page 5, Line 148: Spinal instability is a absolute contraindication to spinal manipulative therapy. Please state which vertebrae were being manipulated and the rationale behind this.
This comment also applies to the use of the cervical Denneroll.
Comment 10:
Table 2: Please state the precise timing of the post intervention x-rays. X-rays taken directly after treatment will show a greater/exaggerated effect size due to ligamentous creep and the "Concertina effect". To avoid this, a window period of at least 2 hours is required before imaging to allow the reader to assess the true effect of this therapy on the cervical lordosis.
Furthermore, some of these measures can be influenced by radiographic positioning. Please indicate who took the pre- and post-imaging and if the patient was set up in the same posture on both occasions. This should be detailed in the methods section.
Comment 11: Statistical Analysis
The paired t-test is not appropriate for analysing the pre- and post-intervention for the right C1 overhang values as the differences are not normally distributed. Please transform and re-analyze, or alternatively use a equivalent non-parametric test.
Please provide the t statistic, degrees of freedom and p-value for each of the paired t-tests conducted. Include the mean difference, standard deviation, and 95% confidence intervals around the mean difference for each outcome of interest.
In analysing data, I am unable to find a significant relationship between the variables mentioned using linear regression. I also do not think that the assumptions for linear regression are met by this particular data set.
I am not able to follow what is portrayed in Figure 3. There is no participant with an 800% change in their ARA value from baseline to post-intervention. The biggest change goes from 3 to 17.9 (Patient 4). This is a 5 fold increase only.
The analysis and results sections need to be revised before further review.
Author Response
Reviewer 1:
We are very grateful for the time and effort of all of the reviewers. Reviewer 1 in particular gave very detailed and helpful suggestions, and we have done our best to incorporate them as follows (responses are in bold brackets []):
My edits and suggestions are provided below:
Comment 1: Overall Manuscript
Please consider reporting this case series in accordance with the CARE guidelines:
https://www.equator-network.org/reporting-guidelines/care/
This will require a change in the way that several sections are written and reported.
[The CARE template has been followed, and is now part of the supplemental material]
Comment 2: Title
Please indicate in the title of the manuscript that this is a case series involving adult patients.
[The title now reads “Non-surgical management of upper cervical instability via improved cervical lordosis: a case series of adult patients.]
Comment 3: Introduction
Please provide the reader with prevalence or incidence figures for both whiplash injury and upper cervical instability. This will allow the reader to gauge the magnitude of the issue. It might also be helpful to highlight the epidemiology regarding how many cases of radiographically appreciable instability are symptomatic. This will also comment on the significance of this work.
[the following sentences, with supporting citations, have been added to the introduction
(line 29)
Injuries resulting from whiplash trauma are common; it is estimated that there are approximately 2.9 million cases of whiplash trauma-associated injury that occur annually in the US. [2]
(line 48)
While the 2.0 mm overhang threshold has only moderate sensitivity and positive predictive value (PPV) for upper cervical injury in the whiplash-injured population with chronic symptoms (64% and 75%, respectively), and the more subjective asymmetry assessment has low sensitivity (29%) but high PPV (95%), in combination the two findings have a PPV of 100%.[9]
Comment 4:
Introduction, Page 2, Line 62. The importance of a normal cervical lordosis needs to be given more support in this section. Or at least, highlight that there are questions that we do not know the answers to at this point in time.
Is there evidence to suggest that it is clinically worthwhile restoring a theoretically ideal posture? Are the benefits worth the cost/effort? Asymptomatic patients often display a loss of the normal cervical lordosis, so is having a normal cervical lordosis essential from a clinical perspective?
[The following has been added at line 76:
The clinical benefits of a lordotic cervical curve have been demonstrated in multiple studies. As an example, in a study of 300 neck pain patients under the age of 40, Gao and colleagues found an increased degree of disk herniation in the patients with straight and kyphotic cervical spines, in comparison with the lordotic necks.[18] They also reported an improvement in disk height and a decrease in disk herniation severity and associated spinal cord compression in the patients who had an improvement in lordosis. A recent systematic review of controlled clinical trials of lordosis restoration therapy demonstrated that symptomatic improvements were maintained for up to 1.5 years after cessation of therapy, when treatment included extension traction directed at improvement of the lordotic curve.[19] In comparison, the control treatment groups were more likely to relapse after cessation of therapy. ]
Comment 5:
Introduction, Page 2, Line 72. CPB should read CBP.
[Fixed]
Comment 6:
Materials and Methods
Page 2, Line 92. Please report median and interquartile range instead of mean for the treatment times as these data are right (positively) skewed.
[Thanks for this recommendation - the median and IQR have been added for treatment duration, beginning on line 116:
The patient histories and treatment course varied widely, and median time between baseline and follow-up radiographic examination was 16 weeks (with an interquartile range of 32 weeks).]
Comment 7:
Page 3 Line 98. 'Typo: less' instead of 'les'
[Fixed]
Comment 8:
Have these patient's been referred for an orthopaedic assessment given that they have displayed radiographic and symptomatic instability in the upper cervical spine?
[the following has been added at line 113:
A finding of fracture, dislocation, or myelopathy or other concerning neurological manifestation of the instability was an exclusion criterion, as such patients would be uniformly referred for urgent neurosurgical or orthopedic evaluation as part of the clinic protocol.]
Comment 9:
Page 5, Line 148: Spinal instability is a absolute contraindication to spinal manipulative therapy. Please state which vertebrae were being manipulated and the rationale behind this.
This comment also applies to the use of the cervical Denneroll.
[The following language has been added:
(Line 217) The manipulations were solely directed at hypomobile spinal segments in the mid and lower cervical spine, as manipulation at the unstable upper cervical spine would be contraindicated.
(line 226) …used like a pillow while the patient is supine, and positioned at the mid to lower cervical spine.]
Comment 10:
Table 2: Please state the precise timing of the post intervention x-rays. X-rays taken directly after treatment will show a greater/exaggerated effect size due to ligamentous creep and the "Concertina effect". To avoid this, a window period of at least 2 hours is required before imaging to allow the reader to assess the true effect of this therapy on the cervical lordosis.
[this is an important point, and one that was considered in the original clinical approach, but not mentioned in the manuscript. The following has been added
Lines 280-1:
Radiographic re-evaluations, performed at least 72 hours after the most recent therapy,
and at Line 137:
Both examinations were performed at the baseline and prior to initiation of therapy, and then repeated no less than 72 hours after therapy was concluded, as the goal was to avoid imaging of any temporary cervical curve improvement directly following extension traction. ]
Furthermore, some of these measures can be influenced by radiographic positioning. Please indicate who took the pre- and post-imaging and if the patient was set up in the same posture on both occasions. This should be detailed in the methods section.
[The following has been added at line 138 to 141:
In order produce images that were consistent with each other, the patient was positioned in the same fashion in both studies, each conducted by the same author, (either EAK or SBK), with the central ray at C5, back or shoulder touching the image intensifier (depending on view), and with a 20 mm marker on the patient’s skin for calibration of the PostureRay® measuring software.]
Comment 11: Statistical Analysis
The paired t-test is not appropriate for analysing the pre- and post-intervention for the right C1 overhang values as the differences are not normally distributed. Please transform and re-analyze, or alternatively use a equivalent non-parametric test.
Please provide the t statistic, degrees of freedom and p-value for each of the paired t-tests conducted. Include the mean difference, standard deviation, and 95% confidence intervals around the mean difference for each outcome of interest.
In analysing data, I am unable to find a significant relationship between the variables mentioned using linear regression. I also do not think that the assumptions for linear regression are met by this particular data set.
I am not able to follow what is portrayed in Figure 3. There is no participant with an 800% change in their ARA value from baseline to post-intervention. The biggest change goes from 3 to 17.9 (Patient 4). This is a 5 fold increase only.
The analysis and results sections need to be revised before further review.
[Thank you for these detailed comments. The statistical analyses have been completely revised, and the data in Table 2 were corrected.
We have added Table 3, which shows the tests for normality. Right C1-2 overhang differences were not normally distributed and are now assessed with the non-parametric signed rank test.
Table 3 includes all of the requested statistics and metrics.
Lines 274+ now reads as follows:
Average treatment effect was assessed via the difference between the pre- and post-treatment radiographic measurements of ARA and lateral mass overhang using the Student’s t-test for normally distributed differences and the signed rank test for non-normally distributed differences. Linear regression was used to assess the correlation between the percent change in ARA and the average of the left and right percent changes in C1-C2 overhang measurement (percent change = [post-measurement – pre-measurement]/pre-measurement). Normality of each difference and the average overhang percent change was assessed using the Shapiro-Wilk test. P-values<0.05 were considered significant. All analyses were performed using SAS Software, version 9.4 (SAS Institute, Inc., Cary, NC, USA).]
Lines 300+ from results now read as follows:
Mean ARA value at baseline was -10°, and -21° after intervention (p = 0.002, see Table 3). Three of the patients (1, 6, and 9) had ARA values at or approaching -30° (see Table 2). There was an average reduction in C1-C2 lateral mass overhang from 6.0 mm to 3.0 mm on the left (p = <0.001), and from 3.1 mm to 1.6 mm on the right (p = 0.004) (see Table 3). The average percent change in C1-2 overhang was normally distributed (p = 0.91). The percent change in ARA and average percent change in C1-2 overhang were moderately correlated (see Figure 3, R2 = 0.46, p=0.04).
Reviewer 2 Report
The research presented in this article suggests that enhancing cervical lordosis can contribute to improvement in signs and symptoms of upper cervical instability secondary to traumatic injury.
A relationship between whiplash and cervical lordosis has never been studied. However, it is very possible that symptoms resolved spontaneously, and that the improvements were unrelated to the treatment or radiographic changes.
The author can also compare the results of patients who did not have improvement in the radiological parameters.
Author Response
Reviewer 2
We thank Reviewer 2 for the helpful comments. In accordance with the suggestions, the following changes have been made, in []s:
The research presented in this article suggests that enhancing cervical lordosis can contribute to improvement in signs and symptoms of upper cervical instability secondary to traumatic injury.
A relationship between whiplash and cervical lordosis has never been studied. However, it is very possible that symptoms resolved spontaneously, and that the improvements were unrelated to the treatment or radiographic changes.
[We have added the following discussion and citation, starting at line 70:
Loss of the normal cervical lordotic curve is a common radiographic finding in patients with chronic pain after whiplash, although there is no general consensus in the literature as to whether the finding indicates true pathology or a normal variant.[16–19]It is well established, however, that the normal cervical lordosis is the biomechanically ideal posture of the cervical spine, as mechanical stresses in the spine are most evenly balanced between the intervertebral disk and zygapophyseal joints when the “C”-shaped curve of the neck is maintained.[20] The clinical benefits of a lordotic cervical curve has been demonstrated in multiple studies. As an example, in a study of 300 neck pain patients under the age of 40, Gao and colleagues found an increased degree of disk herniation in the patients with straight and kyphotic cervical spines, in comparison with the lordotic necks.[21]They also reported an improvement in disk height and a decrease in disk herniation severity and associated spinal cord compression in the patients who had an improvement in lordosis. A recent systematic review of controlled clinical trials of lordosis restoration therapy demonstrated that symptomatic improvements were maintained for up to 1.5 years after cessation of therapy, when treatment included extension traction directed at improvement of the lordotic curve.[22] In comparison, the control treatment groups were more likely to relapse after cessation of therapy.]
The author can also compare the results of patients who did not have improvement in the radiological parameters
[We added the following to the discussion section:
Starting at line 325: The positive changes observed in the imaging is thus much more likely to be explained by the therapy, than the natural course of the condition, which had reached a static level in all of the patients. Ideally, the study would have included a group of patients who did not improve radiographically in either cervical curve or upper cervical instability for comparison with the patients in the case series, and this is a potential future avenue of investigation. ]
Reviewer 3 Report
Thank you for the opportunity to review this interesting paper, exploring a novel approach to the management of cervical instability.
This paper investigates the impact of treatment aimed at influencing the lordosis of the cervical spine on radiologically observed cervical instability in whiplash patients. The premise of spinal shape influencing biomechanics at an inter-vertebral level is a fascinating area, and the aims of the study are clinically valuable. There are concerns however over the generalizability of the results, and some elements of the current study design, that may have resulted in an degree of study bias.
The manuscript is generally clear and concise, presented in a well-structured manner and includes numerous relevant references. There are some concerns however relating to the experimental design, being appropriate to answer the study's aims. The below outlines some of these.
As it stands, it would be very difficult to generalize the results with any certainty to a particular population. This may be better framed as a ‘pilot study’ due to the small sample size. Indeed, some literature would suggest that even for a pilot study, a minimum of 12 participants would be recommended. There is also a lack of distinction in terms of inclusion and exclusion criteria.
‘Patient histories and treatment course varied widely’ with treatment duration ranging from 7 weeks to well over 1 year. There could be more detail concerning exactly when after therapy was concluded the follow-up images were taken. As it stands, if the images are taken directly after treatment, there may be a temporary change in biomechanics. Indeed, it would be useful for there to be more detailed explanation of how and when the images were taken. This should be a highly standardised process in order to remove as optimally as possible, any bias in the results due to patient positioning. The manuscript would benefit from this being explained in more detail.
Another area that requires further detail concerns the non-radiological outcome measures. Regarding the outlined 'Pain scores and Quality of life outcomes', there is no mention of the outcomes used. These would ideally be validated outcome measures. The results in this section look at numerous outcomes, but do not define how any of them, or how they were collected.
Terminology of ‘improved lordosis’ may be best replaced with ‘increased lordosis’. As the manuscript alludes, the literature is scarce when relating spinal curvature specifically to pathology. As the current spectrum of ‘normal’ range of lordosis is relatively large, this heterogeneity means that it is difficult to be sure that changes in either direction are assuredly positive or negative. This could be explored in the discussion.
In terms of the key take home message, explanations of how changing the mechanics of the mid to lower cervical spine may feasibly influence the upper cervical spine appear somewhat speculative. These arguments could be further supported with the use of even more expansive literature, that investigates the inter-dependence of spinal regions during movement.
From an ethics perspective, presumably an ethical review was submitted before applying hyperextension interventions to the cervical spine, as part of a patient-based study. Please refer to this in the manuscript.
Whilst this is an extremely interesting area, in its current state, there are numerous elements of the study design and methodology that could have led potential bias in the findings. As such, the conclusions reached may not be sufficiently supported by the work. It is this reviewer’s opinion that the limitations discussed should be discussed in greater detail, and that the strength of the conclusions are a reflection of them.
Author Response
Reviewer 3
We appreciate the comments and suggestion by Reviewer 3, and have done our best to respond below, in []s.
Thank you for the opportunity to review this interesting paper, exploring a novel approach to the management of cervical instability.
This paper investigates the impact of treatment aimed at influencing the lordosis of the cervical spine on radiologically observed cervical instability in whiplash patients. The premise of spinal shape influencing biomechanics at an inter-vertebral level is a fascinating area, and the aims of the study are clinically valuable. There are concerns however over the generalizability of the results, and some elements of the current study design, that may have resulted in an degree of study bias.
The manuscript is generally clear and concise, presented in a well-structured manner and includes numerous relevant references. There are some concerns however relating to the experimental design, being appropriate to answer the study's aims. The below outlines some of these.
As it stands, it would be very difficult to generalize the results with any certainty to a particular population. This may be better framed as a ‘pilot study’ due to the small sample size. Indeed, some literature would suggest that even for a pilot study, a minimum of 12 participants would be recommended. There is also a lack of distinction in terms of inclusion and exclusion criteria.
[We have added, at line 328, the following:
Because the design of the present study was conceived of only after the association between cervical curve improvement and decreased upper cervical instability was noted, the evidence for symptomatic improvement was only derived from narrative histories, rather than consistently used metrics. Future investigation would thus benefit from an a priori design with standardized objective measurements of the non-radiographic changes described in this study, as well as the inclusion of a comparison group of patients who did not improve radiographically in either cervical curve or upper cervical instability. Moreover, the ability to generalize from this small sample of highly selected patients is limited, and thus another goal of future study is to increase the number of study subjects.]
‘Patient histories and treatment course varied widely’ with treatment duration ranging from 7 weeks to well over 1 year. There could be more detail concerning exactly when after therapy was concluded the follow-up images were taken. As it stands, if the images are taken directly after treatment, there may be a temporary change in biomechanics. Indeed, it would be useful for there to be more detailed explanation of how and when the images were taken. This should be a highly standardised process in order to remove as optimally as possible, any bias in the results due to patient positioning. The manuscript would benefit from this being explained in more detail.
[Added at line 137:
Both examinations were performed at the baseline and prior to initiation of therapy, and then repeated no less than 72 hours after therapy was concluded, as the goal was to avoid imaging of any temporary cervical curve improvement directly following extension traction.
In order produce images that were consistent with each other, the patient was positioned in the same fashion in both studies, each conducted by the same author, (either EAK or SBK), with the central ray at C5, back or shoulder touching the image intensifier (depending on view), and with a 20 mm marker on the patient’s skin for calibration of the PostureRay® measuring software.]
Another area that requires further detail concerns the non-radiological outcome measures. Regarding the outlined 'Pain scores and Quality of life outcomes', there is no mention of the outcomes used. These would ideally be validated outcome measures. The results in this section look at numerous outcomes, but do not define how any of them, or how they were collected.
[please see above, starting at line 328]
Terminology of ‘improved lordosis’ may be best replaced with ‘increased lordosis’. As the manuscript alludes, the literature is scarce when relating spinal curvature specifically to pathology. As the current spectrum of ‘normal’ range of lordosis is relatively large, this heterogeneity means that it is difficult to be sure that changes in either direction are assuredly positive or negative. This could be explored in the discussion.
[Added, at line 290:
Radiographic re-evaluations, performed at least 72 hours after the most recent therapy, revealed substantially improved cervical lordosis (i.e., progress toward the ideal ARA of -34°) in all of the patients.]
In terms of the key take home message, explanations of how changing the mechanics of the mid to lower cervical spine may feasibly influence the upper cervical spine appear somewhat speculative. These arguments could be further supported with the use of even more expansive literature, that investigates the inter-dependence of spinal regions during movement.
[Added, at line 340:
We hypothesize that improvement of the cervical lordosis results in improved biomechanics of the upper cervical spine, and that this in turn allows for improvement of the integrity of the ligaments responsible for craniocervical stability. This hypothesis is an extension of the findings of prior authors, who have described a correlation between increased angle of the upper cervical (C0-2) spine and increased risk of cervical kyphosis.[27] Ours is the first study to demonstrate a relationship between loss of normal cervical curve and symptomatic instability, however.]
From an ethics perspective, presumably an ethical review was submitted before applying hyperextension interventions to the cervical spine, as part of a patient-based study. Please refer to this in the manuscript.
[as the study was conducted retrospectively and on anonymized data, it was exempt from such ethical review or oversight. We have added the following statement to clarify at line 354:
Informed consent was obtained from all subjects involved in the study. As the data were gathered retrospectively from anonymized files of patients who were treated under standard clinical protocol, the study was exempt from institutional review board oversight.]
Whilst this is an extremely interesting area, in its current state, there are numerous elements of the study design and methodology that could have led potential bias in the findings. As such, the conclusions reached may not be sufficiently supported by the work. It is this reviewer’s opinion that the limitations discussed should be discussed in greater detail, and that the strength of the conclusions are a reflection of them.
[Added, at line 347:
Because the design of the present study was conceived of only after the association between cervical curve improvement and decreased upper cervical instability was noted, the evidence for symptomatic improvement was only derived from narrative histories, rather than consistently used metrics. Future investigation would thus benefit from an a priori design with standardized objective measurements of the non-radiographic changes described in this study, as well as the inclusion of a comparison group of patients who did not improve radiographically in either cervical curve or upper cervical instability. Moreover, the ability to generalize from this small sample of highly selected patients is limited, and thus another goal of future study is to increase the number of study subjects.]
Round 2
Reviewer 1 Report
Thank you for the opportunity to re-review this work.
My comments and suggestions are detailed below:
Comment 1:
Page 3, line 127. The paper by Gao et al is a cross-sectional study design. This study design can provide evidence of association only. This paper cannot be used to provide evidence of a 'clinical benefit' (i.e. a causal association between improving cervical lordosis and symptoms) as the authors have suggested.
Comment 2:
Page 3, line 132. The statement from the authors of this SLR, that symptomatic improvements were maintained up to 1.5 years is not possible given that the longest follow-up time in any of the included studies was 15.5 months (Table 1).
Comment 3:
Table 3: 'Mean difference test valueb' in the ARA, and Left overhang columns is the value of the t-statistic for the two sided paired t-test of the differences of the pre and post ARA values and pre and post left overhang values. Not the mean difference.
Comment 4:
Table 3: The results for the Wilcoxon signed rank test (paired) for the pre- and post-differences in right overhang values are incorrect. Please revise.
Comment 5:
I am unable to decipher how you did the linear regression. I also still do not think the data meet the assumptions for linear regression specifically (linear relationship, or multivariate normality). The percent change in ARA is not normally distributed, nor are the percentage change in right overhang values. I would recommend against reporting the results of linear regression in this simple case series.
Author Response
Thank you once again for the careful reading of the manuscript. Our responses are below in bold []s.
Comment 1:
Page 3, line 127. The paper by Gao et al is a cross-sectional study design. This study design can provide evidence of association only. This paper cannot be used to provide evidence of a 'clinical benefit' (i.e. a causal association between improving cervical lordosis and symptoms) as the authors have suggested.
[In the manuscript we wrote the following about this study:
“As an example, in a study of 300 neck pain patients under the age of 40, Gao and colleagues found an increased degree of disk herniation in the patients with straight and kyphotic cervical spines, in comparison with the lordotic necks.[22] They also reported an improvement in disk height and a decrease in disk herniation severity and associated spinal cord compression in the patients who had an improvement in lordosis.”
We do not read the above (in italics) as suggesting that the association was causal. We reported the observations of the authors, but not any of their conclusions regarding the possible clinical benefits of lordosis restoration for this paper]
Comment 2:
Page 3, line 132. The statement from the authors of this SLR, that symptomatic improvements were maintained up to 1.5 years is not possible given that the longest follow-up time in any of the included studies was 15.5 months (Table 1).
[This is a difficult one, as the reviewer is critiquing the original SR authors’ interpretation (and presumed rounding) of their reviewed literature, rather than our interpretation (the 1.5 years was from the study, and mentioned in the abstract), however, we take the reviewer’s point, and have altered the relevant language to “for more than 1 year” at line 86]
Comment 3:
Table 3: 'Mean difference test valueb' in the ARA, and Left overhang columns is the value of the t-statistic for the two sided paired t-test of the differences of the pre and post ARA values and pre and post left overhang values. Not the mean difference.
[Corrected. It was previously requested that we include the t-statistic value. We have changed the row label to “Test statistic” for clarity.
The mean difference is shown two rows above in the row labeled “Mean difference (standard deviation).”]
Comment 4:
Table 3: The results for the Wilcoxon signed rank test (paired) for the pre- and post-differences in right overhang values are incorrect. Please revise.
[Below is the SAS output for right overhang, which is accurately portrayed in the table (the only difference is we rounded 0.0039 to 0.004 in the table):
|
Tests for Location: Mu0=0 |
||||
|
Test |
Statistic |
p Value |
||
|
Student's t |
t |
-3.05215 |
Pr > |t| |
0.0158 |
|
Sign |
M |
-4.5 |
Pr >= |M| |
0.0039 |
|
Signed Rank |
S |
-22.5 |
Pr >= |S| |
0.0039 |
Note: The reviewer may be using different software than SAS. SAS uses the sum of the ranks of the positive values minus the sum expected under the null hypothesis, (n*(n+1)/4). Here that is 0 – 22.5 = -22.5.
Comment 5:
I am unable to decipher how you did the linear regression. I also still do not think the data meet the assumptions for linear regression specifically (linear relationship, or multivariate normality). The percent change in ARA is not normally distributed, nor are the percentage change in right overhang values. I would recommend against reporting the results of linear regression in this simple case series.
[Please see description beginning on line 277 “Linear regression was used to assess the correlation between the percent change in ARA and the average of the left and right percent changes in C1-C2 overhang measurement (percent change = [post-measurement – pre-measurement]/pre-measurement).”
Percent change, as we describe, was calculated for ARA, and left and right overhang. The average of the left and right overhang percent change was also calculated: (left percent change + right percent change)/2. This average was then regressed on the ARA percent change. We have added a clarification to these methods on line 281 which will hopefully suffice.
As noted in lines 306-307 (and despite appearances), the average percent change was normally distributed. Here is the SAS output:
|
Tests for Normality |
||||
|
Test |
Statistic |
p Value |
||
|
Shapiro-Wilk |
W |
0.971563 |
Pr < W |
0.9078 |
|
Kolmogorov-Smirnov |
D |
0.178291 |
Pr > D |
>0.1500 |
|
Cramer-von Mises |
W-Sq |
0.04454 |
Pr > W-Sq |
>0.2500 |
|
Anderson-Darling |
A-Sq |
0.24279 |
Pr > A-Sq |
>0.2500 |
The residuals from the regression analysis do show a slight deviation from the normality assumption (see below), however, we believe it is not so dramatic as to completely omit the finding.
Reviewer 3 Report
The changes you have made I feel really enhance the paper. Good luck with its dissemination.
Author Response
Thanks very much, and for the review!